# A Unified Airspace Risk Management Framework for UAS Operations †

**Suraj Bijjahalli [1], Alessandro Gardi [2,3], Nichakorn Pongsakornsathien [3], Roberto Sabatini [2,3,*] and Trevor Kistan [3,4]**

1. Australian Centre for Field Robotics, University of Sydney, Camperdown, NSW 2006, Australia; suraj.bijjahalli@sydney.edu.au
2. Department of Aerospace Engineering, Khalifa University of Science and Technology, Abu Dhabi P.O. Box 127788, United Arab Emirates; alessandro.gardi@ku.ac.ae
3. School of Engineering, RMIT University, Bundoora, VIC 3083, Australia; s3679479@student.rmit.edu.au (N.P.); trevor.kistan@thalesgroup.com.au (T.K.)
4. Thales Australia, Melbourne, VIC 3008, Australia
* Correspondence: roberto.sabatini@ku.ac.ae
† This article is an extended version of our conference paper: S. Bijjahalli, A. Gardi, N. Pongsakornsathien, and R. Sabatini, "A Unified Collision Risk Model for Unmanned Aircraft Systems", AIAA/IEEE 40th Digital Avionics Systems Conference, DASC2021, San Antonio, TX, USA, 3–7 October 2021.

**Abstract:** Collision risk modelling has a long history in the aviation industry, with mature models currently utilised for the strategic planning of airspace sectors and air routes. However, the progressive introduction of Unmanned Aircraft Systems (UAS) and other forms of air mobility poses new challenges, compounded by a growing need to address both offline and online operational requirements. To address the associated gaps in the existing airspace risk assessment models, this article proposes a comprehensive risk management framework, which relies on a novel methodology to model UAS collision risk in all classes of airspace. This methodology inherently accounts for the performance of Communication, Navigation and Surveillance (CNS) systems, and, as such, it can be applied to both strategic and tactical operational timeframes. Additionally, the proposed approach can be applied inversely to determine CNS performance requirements given a target value of collision probability. This new risk assessment methodology is based on a rigorous analysis of the CNS error characteristics and transformation of the associated models into the spatial domain to generate a protection volume around each predicted air traffic conflict. Additionally, a methodology to quickly and conservatively evaluate the multi-integral formulation of collision probability is introduced. The validity of the proposed framework is tested using representative CNS performance parameters in two simulation case studies targeting, respectively, a terminal manoeuvring area and an enroute scenario.

**Keywords:** Unmanned Aircraft Systems; UAS; UAS Traffic Management; UTM; collision risk; Air Traffic Management; avionics; airspace risk; Communication, Navigation and Surveillance; CNS; Required Navigation Performance; RNP; Required Communications Performance; RCP; Required Surveillance Performance; RSP; robotics; aerial robotics; navigation; tracking; sense and avoid; detect and avoid

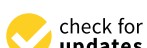



## 1. Introduction

The airspace is currently in a state of transition with regard to the integration of Unmanned Aircraft System (UAS) operations. These operations have largely been segregated from manned aircraft up to this point. However, this is likely to change as more commercial applications and use-cases for UAS emerge. Given the growth rate of the UAS population, the current management framework would not be able to scale to meet the demand efficiently. It is desirable that a highly automated management infrastructure be put in place to holistically coordinate operations between all agents within the shared airspace [1]. The

UAS Traffic Management (UTM) concept was introduced to fill this gap. At a fundamental level, the UTM concept entails a cooperative network that is separate from but complementary to conventional Air Traffic Control (ATC) separation services, supporting the sharing of information on flight intent and airspace constraints between all participating vehicles. Initially, the UTM purview will be restricted to low-altitude uncontrolled airspace below 400 ft. However, there would be growing coordination and interoperability with ATC to support unmanned operations that transition between uncontrolled-controlled airspace and operations above 400 ft [1]. This is illustrated in Figure 1, which shows the current airspace structure. In terms of unmanned traffic, the upper airspace layers are restricted primarily to large, high-altitude (typically defence) platforms conducting surveillance exercises.

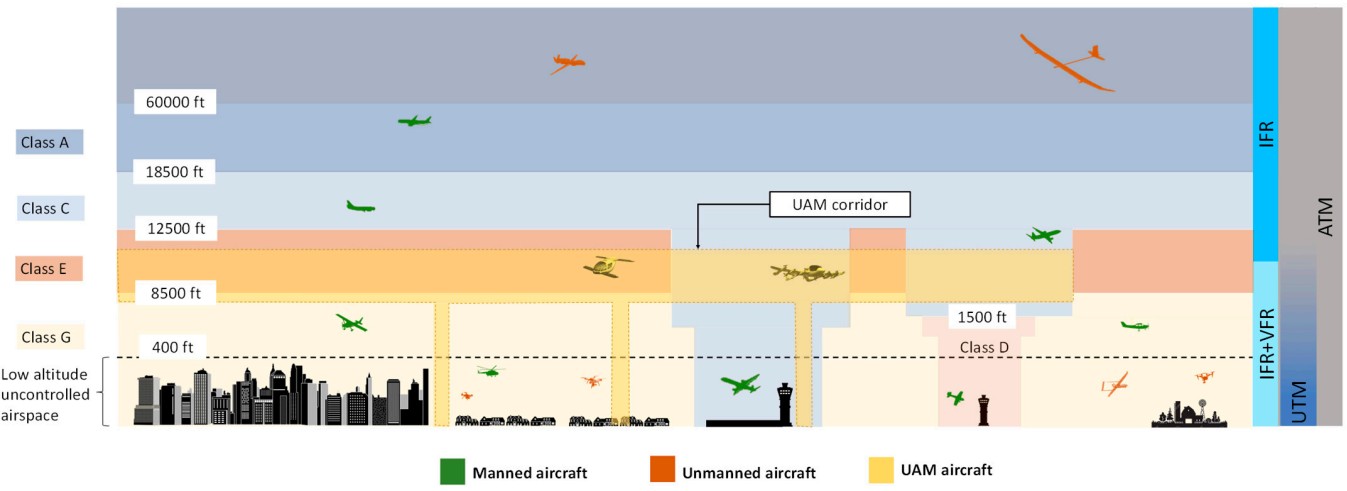

**Figure 1.** Airspace structure incorporating unmanned traffic (note: altitude references are country-specific).

A mature UTM CONOPS will define the responsibilities of both Remote Pilots in Command (RPIC) and UTM operators and will determine the performance envelope within which the variety of UAS vehicles, UTM and ATM systems can implement automation and autonomy. Such UTM CONOPS, however, cannot prescind from the airspace model adopted by the regulator, which shall therefore be evolved coherently with the UTM CONOPS and support it. The UTM architecture envisaged in the latest iteration of the FAA CONOPS for UAS is detailed in [1]. The UTM project is structured into four distinct phases or Technical Capability Levels (TCLs). These phases are characterised by successive progression from Visual Line Of Sight (VLOS) operations in scarcely populated areas to Beyond Visual Line Of Sight (BVLOS) operations in urban areas. The progression of the project is characterised by increasing scenario complexity and required autonomous capabilities [2,3]. Each new TCL extends the capabilities of the previous TCL, with each successive phase supporting a large range of UAS from remotely piloted vehicles to fully autonomous UAS. Each capability is targeted to specific types of applications, geographical areas, and use cases. TCL4 is characterised by complex operations in densely populated urban areas, requiring a number of demanding capabilities. These include Beyond Line Of Sight (BLOS) operations, large-scale contingency management, in-flight deconfliction and trajectory conformance monitoring. In addition to widespread UAS operations, the emergence of on-demand Urban Air Mobility (UAM) services will also place increasing demands on the airspace. UAM and UTM are therefore seen both as subsets of the overarching Advanced Air Mobility (AAM) concept. While UTM is focused on managing unmanned aircraft operations, UAM aims to enable highly automated, cooperative passenger transport in urban and suburban areas. The realisation of this concept is supported by the development of Distributed Electric Propulsion (DEP) and electric Vertical Take-Off and Landing (eVTOL) aircraft, which are envisaged to operate in designated corridors [4]. A UAM corridor is also denoted in Figure 1, spanning multiple airspace classes. As

UAM and UTM operations scale in size, transitions of conventional air traffic and unmanned aircraft across UAM corridors will have to be considered. A highly automated information bridge for communication between ATC, UTM and UAM is called for. To minimise required ATC support in low-altitude airspace, the AAM initiative has proposed the implementation of corridors that can be periodically redefined depending on traffic demand [5,6]. It is expected that inter-aircraft cooperative communication protocols will be mandated to maintain orderly flow and avoid conflicts within the corridor. As in the case of UTM, the envisaged evolution of UAM is characterised by a progressive increase in automation level, with the human operator gradually taking on a more strategic management role. The full realisation of a mature AAM network with seamlessly integrated UTM and UAM operations requires the implementation of several advanced concepts such as 4-Dimensional (4D) Trajectory-Based Operations (TBO) [7–9], Dynamic Airspace Management (DAM) [10] and Performance-Based Operations (PBO). Each aircraft would essentially be assigned a 4D trajectory that is continuously negotiated, monitored and updated in real-time/quasi-real time. This is fundamentally different from legacy flight plans in manned aircraft operations. The implementation of TBO within the dense route structures and short timeframes characteristic of UTM operations is, in turn, dependent on reliable and high-performance Communication, Navigation and Surveillance (CNS) systems. The applicable performance requirements for these systems are embodied under Performance-Based Communication/Navigation/Surveillance (PBC/PBN/PBS) standards. The application of these standards supports trajectory conformance and safe separation.

DAM supports enhanced Demand Capacity Balancing (DCB) by allowing airspace sectors to be dynamically reconfigured or morphed in response to traffic patterns, weather patterns and other dynamic factors. An algorithmic morphing requires the use of metrics or safety indices that characterise the safety of operations as well as the impact of proposed changes in a specific airspace sector. A number of safety indices have been proposed in the literature. A review of these indices can be found in [11]. Traditionally, the probability of aircraft collision or collision risk has been widely used in conventional ATM processes as a metric to assess the safety of operations and airspace structure design/reclassification. Since air-air/air-ground collisions are the primary hazards associated with unmanned aircraft operations, collision risk is well suited as an assessment metric to drive airspace management in the different timeframes of a UAS operation.

Collision risk is typically measured and specified in units of hazard rates (i.e., number of hazardous events per unit of time). The most commonly applied unit is fatal accidents per flight hour. Therefore, its use in Decision Support Systems to trigger airspace management actions such as trajectory changes or sector morphing allows interventions to be traceable and commensurate to an absolute safety metric. In general, aviation authorities have embraced a risk-based approach in the process of developing new standards, operational procedures and safety assessments for supporting the introduction of UAS in the airspace [12,13]. In [14], a risk assessment framework for UTM was proposed, which took a number of internal (to the aircraft) and external environmental parameters into account to evaluate the risk associated with a mission profile. Each phase of a UAS operation can be assessed through a suitably formulated collision risk model. The relevant operational timeframes are adapted from [15] and are illustrated in Figure 2. During the offline phase, when the aircraft is on the ground, statistical traffic flow predictions at a macroscopic level are made, typically using historical data within individual sectors (tactical) and across multiple sectors (strategic). The assessed risk is mitigated through optimising airspace design and flight schedules to ensure that traffic management resources are adequate to meet the foreseen/nominal demand in the foreseen operational conditions. During the online phase, while the mission is in progress, risk assessment is performed over aircraft intents spanning multiple sectors (strategic) or within one sector (tactical), with the former mostly in the form of DCB interventions, while the latter normally involves tactical deconfliction by Air Traffic Controllers (ATCOs) [10]. As part of the ATM evolutionary roadmap, these will be increasingly performed through TBO and DAM, which will play key roles in UTM

operations due to the high reliance on automation. The emergency timeframe is set based on the time budget (with a buffer) allocated to a remote pilot and "safety net" systems to assess the potential of a collision and resolve the conflict [16].

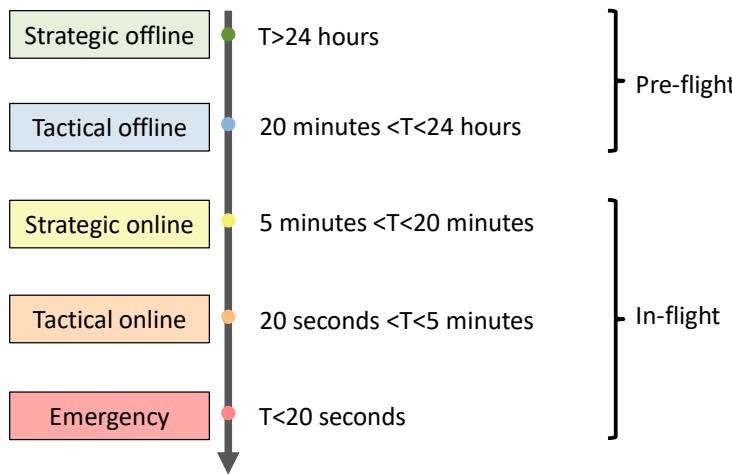

**Figure 2.** Operational timeframes for the envisioned UAS and UAM frameworks.

To be applicable to the UAS and UAM context, a collision risk model must possess several key attributes:

- it should be conservative (since it is always preferable to overestimate risk for a Safety of Life application);
- it should explicitly budget contributions from different CNS elements to the overall risk;
- it should be capable of seamlessly and efficiently handling free routes and arbitrary trajectories.

### 1.1. Scope and Structure of the Article

The key contribution of this paper is a comprehensive, cohesive and robust collision risk assessment framework which encompasses the CNS infrastructure and performance and its ongoing evolutions, along with emergent aspects of the current airspace such as the introduction of UAS and UAM. The framework is also flexible enough to accommodate the dynamics of the aircraft involved in the encounter. Additionally, human factors can also be incorporated into the framework. This includes aspects such as the performance of pilots, ATCOs and other stakeholders. The remainder of Section 1 will review prior work in collision risk modelling methodologies and identify limitations which restrict their application to current and emerging operations. After introducing the fundamental definitions and notation, Section 2 outlines the proposed unified risk assessment framework, beginning with an overview of the methodology followed by its mathematical underpinnings. The application of the proposed framework to various timeframes of operations is also discussed before a demonstration of the same through dedicated case studies in Section 3. Conclusions and recommendations for future research are provided in Section 4.

### 1.2. Prior Work in Collision Risk Modelling

Collision risk modelling is a long-standing field of study for manned aviation. In general, the deployment of new models or proposed updates to existing models occurs at a very slow pace owing to the requirement for extensive verification and validation. As a result, models that were developed as far back as the 1960s are still applied today with modifications and enhancements. The models surveyed for manned aircraft typically occupy one of the broad categories illustrated in Figure 3. The most mature and commonly deployed models in practice are analytical models, either with a closed-form expression or an integral expression that is evaluated numerically. Of these, the Reich-Marks model [17–19]

is the oldest and most widely applied. The model estimates the probability of a mid-air collision between two enroute level flying aircraft. The main objective is the determination of risk associated with lateral separation between adjacent parallel routes or vertical separation between adjacent flight levels. The model applies an ICAO-supported approach and is simple to implement and evaluate. In particular, the model has an extensive history of application in strategic offline scenarios to determine the risk associated with dense routes. The inputs comprise the number of aircraft over an observation period, relative velocities, aircraft dimensions and position error distributions. The model applies a number of simplifying assumptions that make it inapplicable to a broad range of scenarios.

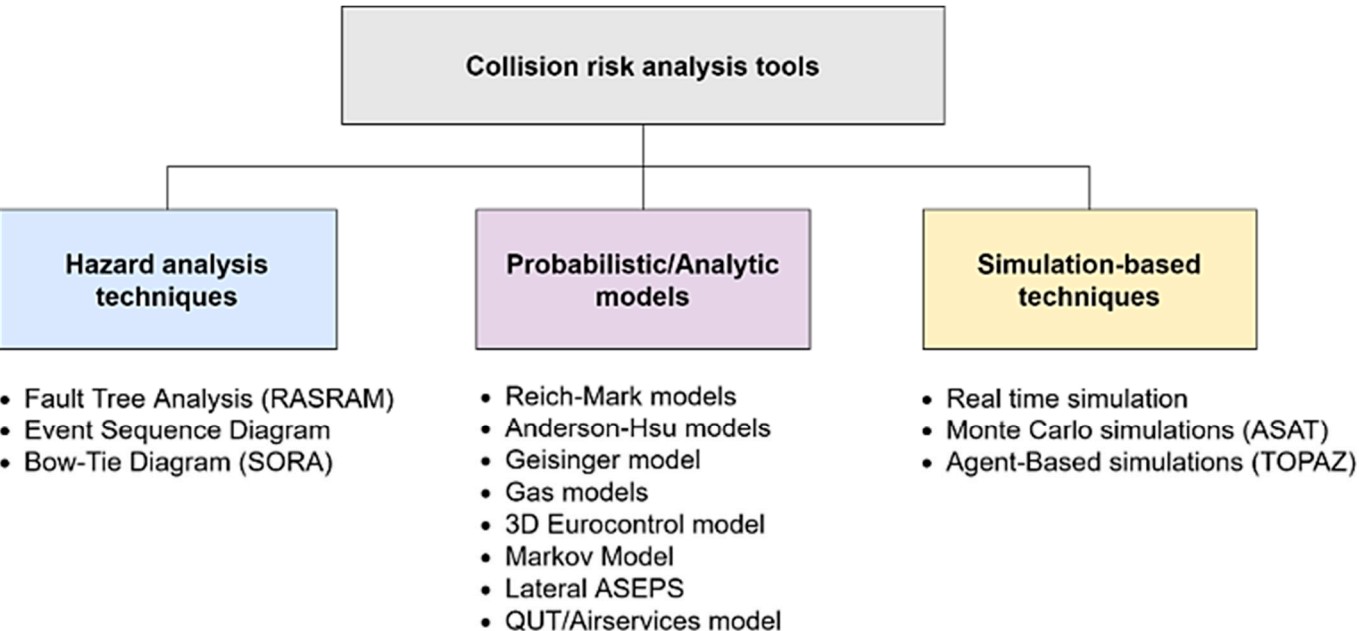

**Figure 3.** Ontology of collision risk models.

The models that followed were in part developed to overcome some of these limitations. For instance, the Anderson-Hsu model and subsequent extensions [20,21] are applicable to aircraft on both parallel and intersecting routes and can accommodate different navigational performances in cross- and along-track directions for both aircraft. The impact of communication and surveillance infrastructure servicing a pair of aircraft is also more readily incorporated in these models. The improved accuracy and applicability, however, come at the expense of computational run time since the method calls for a multiple-integral expression to be evaluated. Other notable examples of analytical models include [22], in which a geometrical model of the aircraft encounter is used in conjunction with empirical distributions characterising pilot and controller reaction times. In [23], nominal aircraft positions were propagated over a tactical timeframe using an assumed kinematic model, and an integral over estimated positions was approximated. In [24], the probability of vertical loss of separation was computed by estimating altitude distributions through a combination of simulation and real data. The impact of wind forecast uncertainty on conflict probability estimation was investigated in [25]. Gas models and their generalisations operate on the principle that aircraft behaviour is random, like the motion of gas molecules in a container. This is useful in obtaining a conservative assessment of risk where there is little to no historical traffic data. ATC environments without surveillance capability, such as see-and-be-seen Visual Flight Rules (VFR) environments that involve mostly General Aviation (GA) aircraft, are a prime candidate for this model. Moreover, surveillance, communication and intervention capability cannot be natively accounted for. Simulation-based techniques apply a markedly different approach to risk modelling. In general, the approach is highly flexible and allows the analysis of a very wide range of

causal and complex factors contributing to a collision. Cascading system and personnel failures can be accounted for, which would be highly complex to capture in an analytical approach. Agent-based simulations and Monte Carlo simulations are prime examples of this. Highly realistic scenarios can be constructed and evaluated. However, improved model fidelity is at the expense of computational expenditure. The Traffic Organization and Perturbation AnalyZer (TOPAZ) model [26] is one such example of a highly flexible framework that employs a combination of Agent-Based Models and Monte Carlo simulations to construct and evaluate complex scenarios. Collision risk modelling for UAS is currently an active area of research owing to the growing number of commercial and recreational operations and the prospect of gradually desegregating the airspace. The field of study is not as mature as in the case of manned aircraft. Nevertheless, several recent models apply methodologies that have been tried and tested on manned aircraft for decades. For example, in [27], the probability of mid-air collision was computed using principles related to the collision frequency of gas molecules as embodied in the gas model. Similarly, in [28], the collision risk between general aviation aircraft and unmanned aircraft was computed using principles from the gas model theory. In [29,30], a data-driven collision risk model was presented, which computed vertical overlap probabilities of manned and unmanned aircraft in the proximity of aerodromes. The model is essentially based on the convolution of altitude error distributions for manned and unmanned aircraft. Error distributions for manned aircraft are approximated through kernel density estimation applied to radar surveillance data. Vertical collision risk is computed and compared, assuming different types of distributions for unmanned aircraft altitude errors. The model itself has not been validated in an operational setting. However, the underpinning mathematical formulation is based on determining the overlap of distributions, which is a well-established methodology. A methodology based on Monte-Carlo sampling to propagate unmanned aircraft trajectories was presented in [31] to evaluate the probability of collision with manned aircraft in restricted airspace scenarios. A large body of UAS research is focused on the Detect-and-Avoid (DAA) problem [32] and on the use of radar or active and passive electro-optical sensors [33] for detecting static and dynamic obstacles [34–37]. In general, most of the proposed DAA solutions do not quantify the risk as a direct function of the probability of collision. Rather, a set of metrics and thresholds are defined that serve as proxy variables for defining a collision. For instance, time- and distance-based metrics (tau, modified tau, DMOD) and thresholds are specified for caution and warning alerts in the Minimum Operational Performance Standards (MOPS) for DAA systems that initiate avoidance manoeuvres [38]. This framework was extended in [39], where a methodology for assuring the integrity and continuity of estimating these metrics was presented. A number of static and dynamic metrics were presented as part of a risk-based framework in [40]. A key finding that emerges from the review of the prior work is that surveillance and communication performance was not inherently accounted for in the early models. Most early models essentially assumed that aircraft were 'flying blind' (i.e., aircraft on a collision course would remain on the same course with no incorporation of intervention mechanisms). Eventually, as in the case of [20], the capability of triggering an intervention based on the communication and surveillance infrastructure was incorporated into several models. This was typically parameterised in terms of the time required to perform an intervention, quantified based on empirical distributions. Most of the models published so far focussed on the offline planning (strategic) timeframe only (i.e., to drive the design of airspace structures and air routes based on historical traffic data) [41]. In the case of unmanned aircraft, mid-air collision risk models have a shorter history of development but typically utilise modelling techniques that have previously been used for manned aircraft. As UAS are expected to have greater interactions with general aviation aircraft operating in uncontrolled airspace over the short to medium term, gas model-based methods are popular since they apply conservative assumptions regarding the behaviour of aircraft in the absence of data. A major gap that emerges from the review is that the majority of the traditional models are largely data-driven [41].This presents problems when assessing the

risk of introducing new aircraft (such as UAS), equipment classes and operations. There is no holistic model-driven approach that accounts for CNS performance that is applicable in both offline planning and real-time operational timeframes.

## 2. Models and Methods

Aircraft collisions are essentially discrete events whose occurrence is uncertain. Probability is one of the most widely accepted quantitative descriptions of such uncertainty. Since a collision can result in loss of life and property, the probability of a collision is also referred to as collision risk. Collision risk has a long history of application in aviation towards the safety assessment of operations and associated equipment, resources, and infrastructure. However, the operational and environmental parameters for manned passenger aircraft are vastly different from those that characterise current (and emerging) UAS operations. As a result, most collision risk models for manned aircraft are inapplicable to UAS. This section provides the context behind the evolution of the airspace due to the introduction of UAS operations and the implications this would have on collision risk modelling and evaluation.

### 2.1. Fundamental Definitions

In a deterministic formulation, a collision event can be defined as the simultaneous occurrence of a horizontal and vertical overlap between aircraft. To evaluate these conditions, aircraft bodies are typically approximated as a regular shape to simplify the development and application of collision risk models. Typical geometries include cylinders and spheres, as illustrated in Figure 4. For computational efficiency, it is common practice to consider one of the aircraft as a point object and inflate the dimensions of the other aircraft so that a critical collision region equivalent to the sum of the individual aircraft dimensions is defined. A collision event then simply corresponds to the point aircraft entering the inflated collision region, as conceptually illustrated in Figure 5.

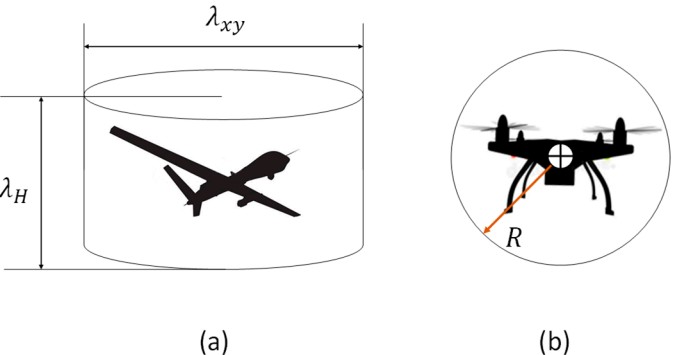

(a)             (b)

**Figure 4.** Conventional representations of aircraft bodies. (**a**) Cylinder of lateral dimension $\lambda_{xy}$ and vertical dimension $\lambda_H$; (**b**) Sphere of radius $R$.

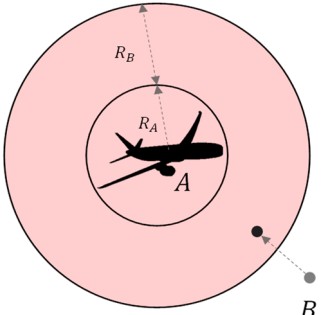

**Figure 5.** Collision definition—breach of a collision region encompassing the combined dimensions of the individual aircraft bodies. Aircraft B enters the collision region volume defined around aircraft A.

A threshold value for the collision risk is typically specified as the so-called Target Level of Safety (TLS), commonly specified in units of fatal accidents per flight hour. A comprehensive review of different TLS values adopted over the years for different operations is provided in [42]. These threshold values typically have a basis in historical accident data.

## 2.2. Unified Airspace Risk Management Framework

A cohesive collision risk assessment framework is introduced henceforth, which encompasses CNS performance and its ongoing evolutions, along with emergent aspects of the current airspace such as the introduction of unmanned aircraft, urban air mobility and space launch and re-entry. The high-level flowchart of this enhanced CNS performance-based framework is depicted in Figure 6. The methodology is tailored to the strategic offline phase of operations, in which historical surveillance or simulated data is used to assess the risk associated with operations over a given region or route. The risk model is nonetheless applicable to online timeframes as well, but this aspect will be covered in future work. For all aircraft pairs in the simulation scenario passing the preliminary traffic filtering step, a multi-step collision risk determination process is carried out. The primary step in this procedure is the generation of a protection volume (i.e., the CNS protection volume around each aircraft involved in the scenario). This volume accounts for errors in navigation and surveillance, delays in executing an intervention due to the limitations of the employed communication system(s), and delays introduced by operator/pilot response times. Additionally, the framework allows for multiple inflations to be added to the volume to compensate for aircraft dynamics and adverse weather conditions. Each of these factors can be mapped to a corresponding inflation of the volume. A penetration of this volume by another aircraft or by a ground obstacle then represents a collision risk. Evaluation of this collision risk and subsequent assessment against a threshold triggers an appropriate strategy for controlling the risk.

The remainder of this section will focus on the collision risk modelling phase in Figure 6 as applied to the strategic offline timeframe of operations. The development of risk control strategies and their deployment in response to assessed risk exceeding an established TLS is beyond the scope of this paper and will be covered in future work.

## 2.3. Collision Risk Mitigation and Control Measures

To ensure that the probability of occurrence does not exceed the assigned threshold, the hazard must either be removed, or the likelihood of risk arising from it must be minimised. To this end, several risk-mitigating measures can be employed for a given scenario, several of which are mentioned at the bottom of Figure 6. A risk mitigation measure is essentially a safety barrier that minimises the probability of occurrence of the collision. A properly designed risk management process requires (1) Detection of the hazard; (2) Assessment of the hazard; (3) Implementation of risk reduction and control actions; (4) Monitoring of satisfactory outcomes. This entire process could be implemented either as a human-in-the-loop system (as a conventional ATM), an autonomous system or a combination of both. Risk reduction and control actions in the strategic deconfliction context would be managed by a UTM system limiting the density of unmanned aircraft to levels where their onboard DAA can safely operate. The approval/denial process may take into account the performance and equipping of the vehicle. A full analysis of the most effective risk reduction and control measures in each operational context and to which these reduce the actual collision risk is beyond the scope of this article and will be addressed in future research.

## 2.4. CNS Performance-Based Collision Risk Modelling

This section presents an overview of the proposed methodology to evaluate the collision risk arising from a given encounter between two or more aircraft. The methodology is a unified approach with the flexibility to account for the performance of the CNS infrastructure servicing the aircraft involved in the encounter. The methodology can be applied in one of two ways: (1) Evaluate risk given a set of operational and system parameters; (2) Solve

the inverse problem (i.e., determine the required CNS performance given a prescribed not-to-exceed TLS). This dual-application capability is illustrated in Figure 7.

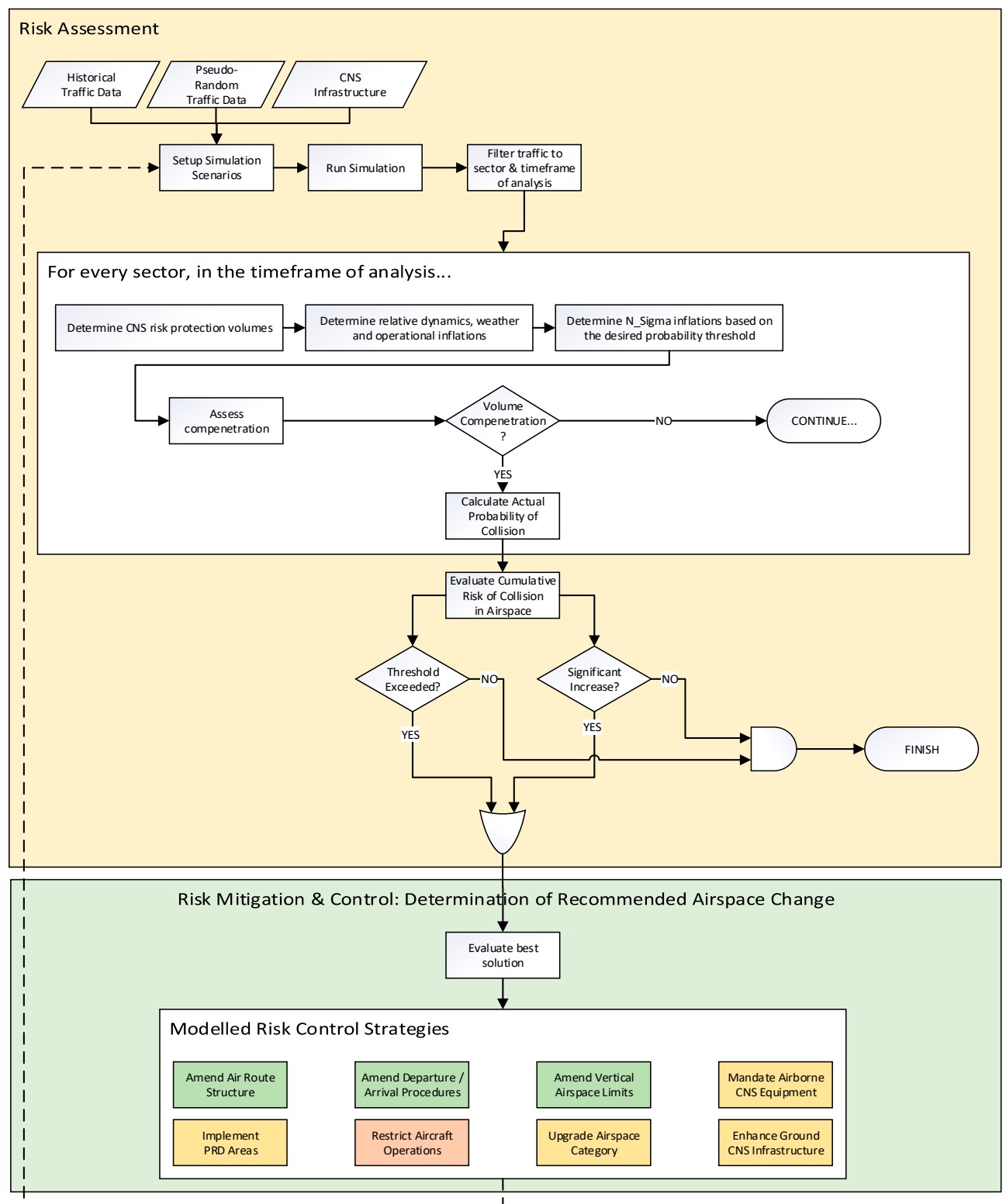

**Figure 6.** Top-level flow-chart of the enhanced CNS-based risk assessment process.

**Forward model application**

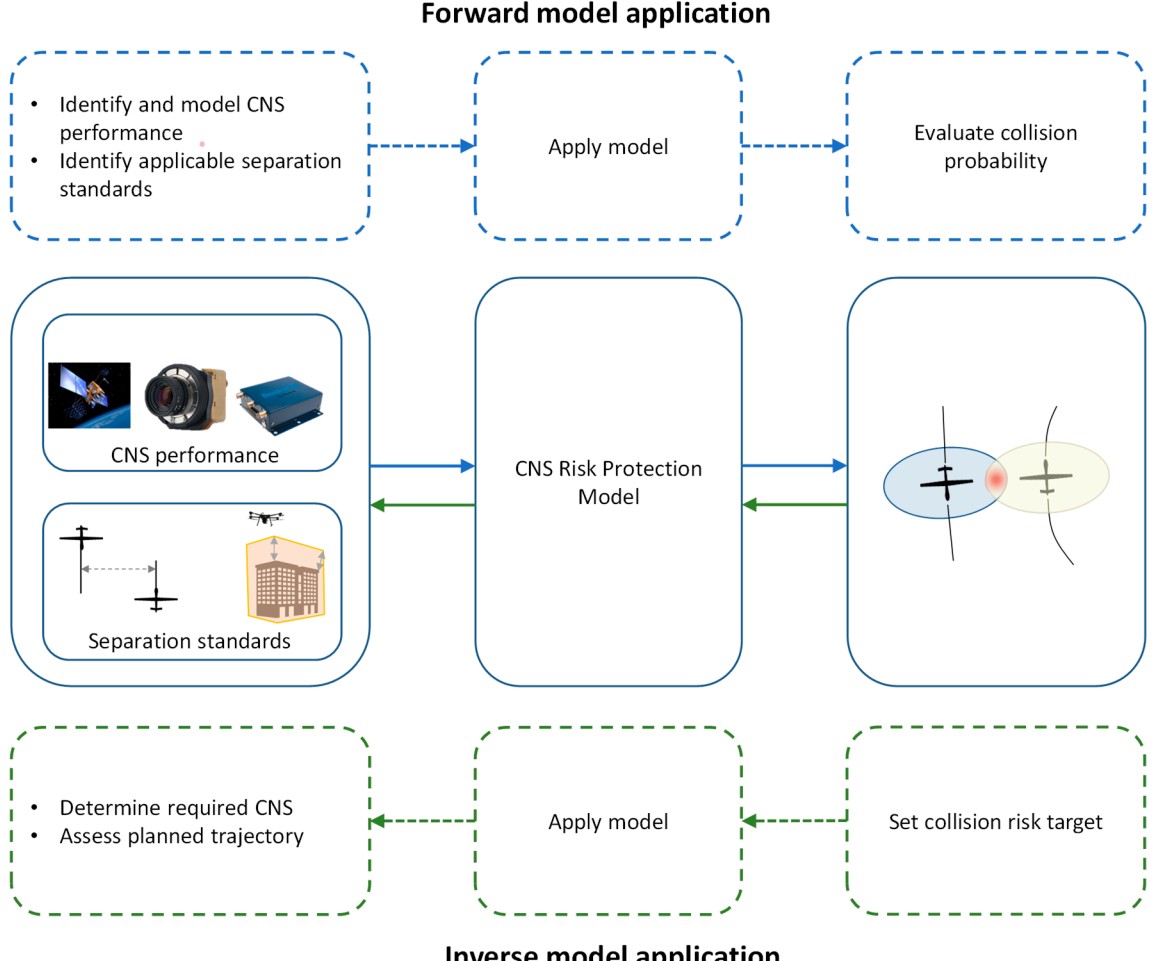

**Inverse model application**

**Figure 7.** Application of the unified approach to evaluate risk or to solve the inverse problem of evaluating performance trade-offs.

The risk evaluation for a proposed mission begins with identifying and modelling the performance of the CNS infrastructure supporting the mission. These serve as inputs to the unified collision risk model along with the planned trajectory and applicable separation standards. The model outputs the probability of collision with other aircraft (given knowledge of their intents over the assessment period) and with terrain features (given a terrain model). In the inverse application, the maximum allowable collision probability is set as a target input to the model, which outputs the trade-space between the required CNS performance and the planned mission profile. The development and assessment of proposed separation standards can be supported in this manner. The high-level procedure in the forward model application (risk evaluation) is illustrated in Figure 8.

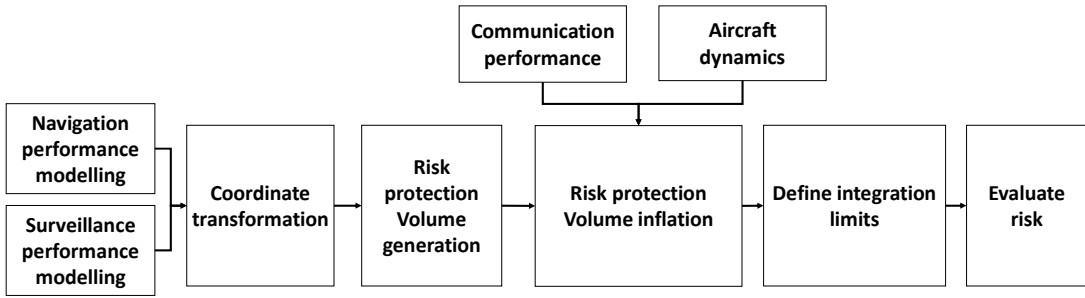

**Figure 8.** Top-level protection volume generation and risk evaluation process.

The methodology is centred on the generation of volumes that bound the nominal positions of the aircraft. These volumes represent the uncertainty in CNS performance and define temporal and spatial boundaries within which the collision risk is to be evaluated.

The first step is the identification of the CNS systems employed by the aircraft and ground infrastructure. The performance of these systems is then modelled and the corresponding Probability Density Functions (PDF) are used to generate bounding volumes around the nominal positions of the aircraft. Additionally, the framework allows for inflations to be added to the volume to compensate for delays in executing an intervention due to the limitations of the employed communication system(s) and delays introduced by operator/pilot response times. Aircraft dynamics and inclement weather conditions can also be mapped to a corresponding inflation of the volume. A penetration of this volume by another aircraft or by a ground obstacle then represents a collision risk. The risk evaluation for a given pair of aircraft is performed by integrating the volume over the conflict/collision region.

### 2.5. Model Formulation

The conventional starting point to formulate a collision risk model is to consider an encounter scenario where one aircraft, designated 'host' or '*aircraft a*' is in the proximity of an 'intruder' aircraft, also designated '*aircraft b*'. This is depicted in Figure 9. The cases of "in trail" aircraft and aircraft flying on parallel routes (both in the same and in opposite directions) are also implicitly encompassed.

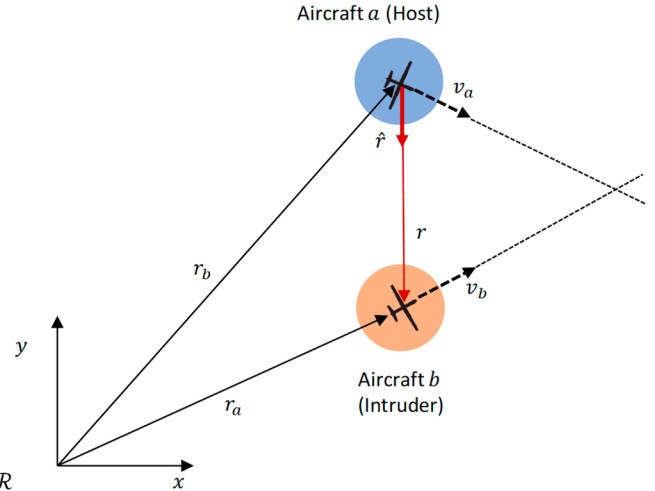

**Figure 9.** Prototypical encounter scenario between two aircraft.

The kinematic states of each aircraft, consisting of individual position and velocity in a reference frame $\mathcal{R}$, are merged for convenience in a global state vector $X_{ab}$:

$$X_{ab} = (r_a, v_a, r_b, v_b)^T \in \mathbb{R}^n \tag{1}$$

The relative motion between aircraft is defined in terms of relative position and velocity, respectively:

$$r = r_b - r_a \tag{2}$$

$$v = v_b - v_a \tag{3}$$

The dynamics of the joint system are therefore expressed as:

$$\dot{X}_{ab}(t) = f(t, X_{ab}(t)), t \in [0, T], \ X_{ab}(t_0) = X_0 \tag{4}$$

where $T$ is a terminal time horizon specifying the duration of the scenario and $X_0$ are the initial conditions. At any given time, a collision occurs if the following condition is met:

$$\| r_b - r_a \|^2 - R^2 < 0 \tag{5}$$

where $R$ is a suitably defined distance threshold, which can be either a scalar value (describing a sphere) or a function of angular or other coordinates (describing a convex three-dimensional shape). Considering that aircraft states are not known deterministically but are instead estimated based on uncertain sensor measurements, it is appropriate to introduce the estimated aircraft states:

$$\hat{X}_{ab} = (\hat{r}_a, \hat{v}_a, \hat{r}_b, \hat{v}_b)^T \in \mathbb{R}^n \tag{6}$$

The collision event becomes a probabilistic one, reflecting the uncertainty in state estimates. To evaluate the probability of the collision event, it is necessary to know or assume the Probability Density Function (PDF) of the relative distance $r$, and then integrate such PDF over a volume which encompasses the distance threshold $R$. Therefore, the instantaneous probability of collision is given by:

$$P_c(t) = \iiint\limits_V p(\boldsymbol{r}) \mathrm{d}\boldsymbol{r} \tag{7}$$

where $p(\boldsymbol{r})$ is the PDF of the relative distance and $V$ is the integration volume which depends only on the shape approximating the aircraft bodies. The first problem in solving Equation (7) is in determining the PDF of the relative position, which in turn depends on the PDFs describing the localisation of the host and intruder aircraft. The second lies in defining the integration limits bounding the volume $V$. If the aircraft bodies are represented by spheres, then the volume $V$ is also a sphere with radius $R = R_a + R_b$ which is the combined aircraft body radius. This is equivalent to stating that a collision event occurs when the relative distance $r = r_b - r_a$ is less than the sum of the two aircraft radii.

$P_c$ can either be evaluated at the current instant or over a time horizon by projecting the aircraft localisation PDFs along the estimated trajectories (obtained by integrating the nominal equations of motion for each aircraft). This is illustrated in Figure 10. The projected Time of Closest Approach (TCA) is of particular interest as an epoch to evaluate the collision probability. This methodology can also be applied in a strategic sense by applying conservative assumptions regarding the expected CNS performance over segments of the planned trajectories. As mentioned, there are two PDFs that must be known or assumed before determining the distribution of the relative distance:

- position of the host aircraft;
- position of the intruder aircraft.

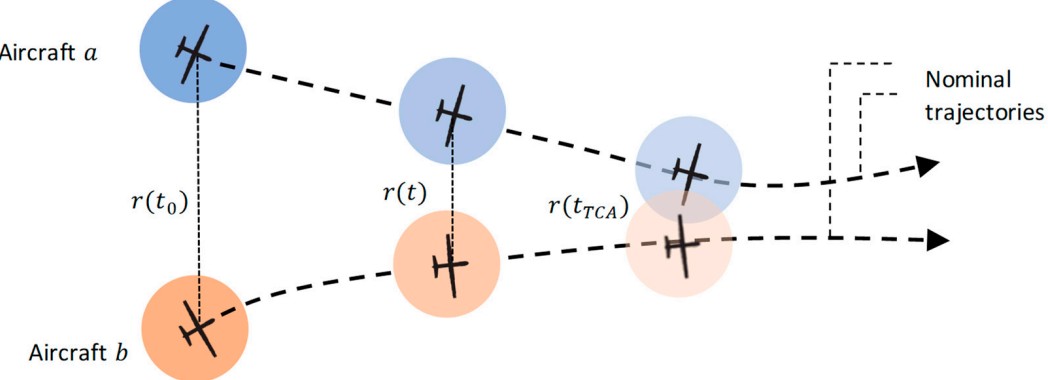

**Figure 10.** Projection and evaluation of collision probability over a time horizon.

In principle, these variables can be typically assumed to follow a Gaussian distribution, as this is the de-facto standard for navigation and surveillance systems in aviation, providing a relatively simple means of characterising uncertainty in system states.

For completeness, the basic formulation of a multi-variate Gaussian probability density function is presented here. Let $\mathcal{N}_n(\xi, \eta, P)$ represent a normal distribution for an $n$-dimensional variable $\xi$ with mean $\eta$ and covariance matrix $P$. Thus, we have:

$$\mathcal{N}_n(\xi, \eta, P) = \frac{1}{\sqrt{(2\pi)^n}} \frac{1}{\sqrt{|P|}} exp\left[-\frac{1}{2}(\xi - \eta)^T P^{-1}(\xi - \eta)\right] \tag{8}$$

As an illustrative example, Figure 11 shows aircraft positioning error modelled as a Gaussian random variable in three dimensions. The ellipsoid visible in such a figure is the 3D representation of the equiprobability surface for a purely Gaussian distribution, which is the most convenient graphical representation. The dimensions and orientation of the ellipsoid are dictated by the elements of the covariance matrix $P$ or their multiples (for instance leading to 1-, 2- or 3-σ ellipsoids, among others).

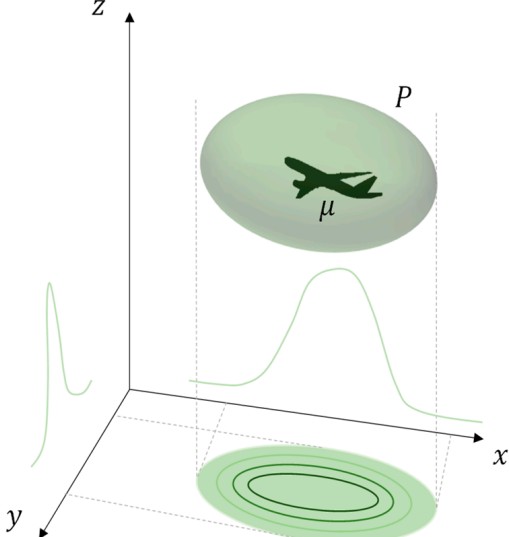

**Figure 11.** The Gaussian ellipsoid represents the uncertainty in aircraft positioning.

Under the Gaussian assumption, the position of each aircraft can be described simply as a function of the nominal position and covariance as:

$$p_a(r_a, t; t_0) = \mathcal{N}_3(r_a, \mu_a, P_a) \tag{9}$$

$$p_b(r_b, t; t_0) = \mathcal{N}_3(r_b, \mu_b, P_b) \tag{10}$$

Assuming the collision risk to be evaluated around the host aircraft, $p_a(r_a)$ represents the error in the navigation system on board aircraft $a$ and $p_b(r_b)$ represents the error in the surveillance system which is deployed at a given epoch to monitor aircraft $b$. The relative distance is simply the distance vector between the two platforms, which is the difference between two Gaussians as illustrated in Figure 12:

$$p(r) = \mathcal{N}_3(r_b, \mu_b, P_b) - \mathcal{N}_3(r_a, \mu_a, P_a) \tag{11}$$

This is equivalent to a convolution operation:

$$\mathcal{N}_3(r_b, \mu_b, P_b) - \mathcal{N}_3(r_a, \mu_a, P_a) = \mathcal{N}_3(r_b, \mu_b, P_b) \oplus \mathcal{N}_3(r_a, \mu_a, P_a) = \mathcal{N}_3(r, \mu, P) \tag{12}$$

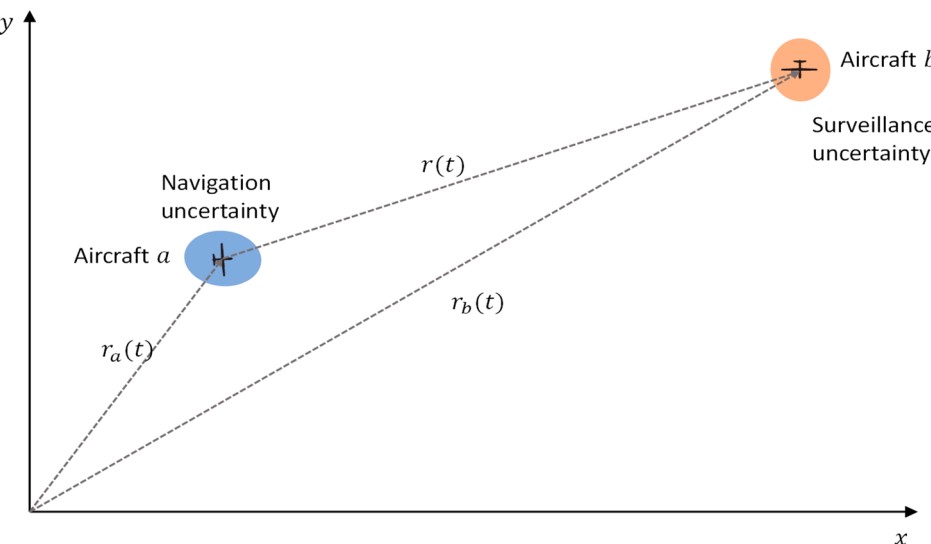

**Figure 12.** Relative distance as a function of host navigation and intruder surveillance uncertainty.

The resulting distribution of the relative distance is also a Gaussian mean:

$$\mu = \mu_b - \mu_a \tag{13}$$

and covariance matrix:

$$P = P_a + P_b \tag{14}$$

where $P$ now represents the combined uncertainty in navigation and surveillance estimates. This uncertainty volume is placed at the nominal position of the intruder aircraft. The probability of collision is essentially the influx of the relative position probability distribution into the collision region defined by the sum of radii of the two aircraft. The collision region is placed at the nominal host position and defines the integration volume $V$ to be evaluated. This is illustrated in Figure 13.

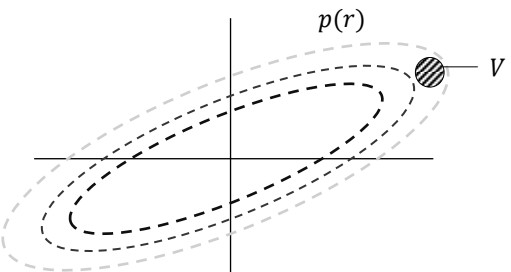

**Figure 13.** Relative distance uncertainty and integration volume. Contours of equal probability are shown.

The probability of collision can now be expressed as a three-dimensional integration of the relative distance over the spherical integral volume $V$:

$$P_c = \frac{1}{(2\pi)^{3/2}|P|^{1/2}} \iiint\limits_V \exp\left(-\frac{1}{2}r^{\mathrm{T}}P^{-1}r\right)dxdydz \tag{15}$$

An exact closed-form solution of Equation (15) is not available for any of the shapes typically adopted in collision risk modelling. Therefore, such an equation must be numerically integrated or, alternatively, a Monte Carlo simulation-based approach can be adopted. Both options present practical issues for real-time implementation owing to their computational costs. Even for an offline evaluation, Monte Carlo simulations are impractical since

the probability of collision is typically an extremely small value and evaluating it through this approach would require a prohibitive number of samples.

### 2.6. Risk Evaluation

Here, a methodology is adopted for approximating the spatial triple integral for collision risk (Equation (15)) conservatively and in closed form, thus overcoming the previously mentioned difficulty of efficiently and accurately evaluating it. In particular, by approximating the volume integral domain as a suitably orientated overbounding cuboid, we can apply the Cumulative Distribution Functions (CDF) along each dimension. Following this approach, the integral evaluation is replaced by an algebraic expression which is immediate to compute and can be inverted to determine trade-offs in system performance. The approach is illustrated in Figure 14, which for simplicity, shows a 2D ellipse.

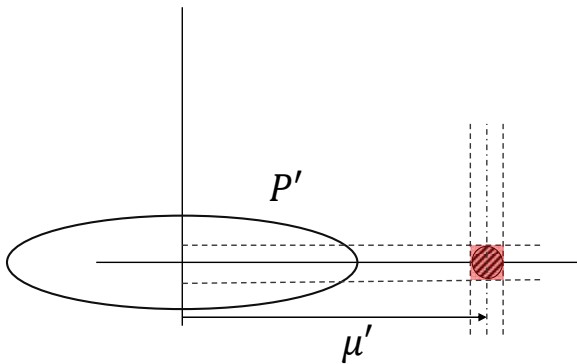

**Figure 14.** Risk evaluation methodology. The rotated integral domain is approximated by a cube (highlighted in red).

First, a rotational transformation is applied to the relative distance distribution. A transformation method that is commonly employed [23] is adapted here. This is accomplished by first performing an eigenvalue decomposition of the generic protection volume $P$:

$$PY = Y\lambda \tag{16}$$

where $Y$ is the matrix of eigenvectors, and $\lambda$ is a diagonal matrix of corresponding eigenvalues. Rewriting Equation (16) as:

$$P = Y\lambda Y^{-1} \tag{17}$$

Equation (17) essentially recasts $P$ as the combination of a rotation and scaling operation. $Y$ represents the orientation of the volume and $\sqrt{\lambda}$ represents a set of scaling parameters along each axis. Applying the rotation yields:

$$P' = Y^{\mathrm{T}}\lambda\,Y \tag{18}$$

By rotating the volume $P$, the integration domain is shifted to:

$$\mu' = Y^{\mathrm{T}}\mu \tag{19}$$

The evaluation of the collision risk integral is therefore rewritten as the difference between Cumulative Distribution Functions (CDF) $\Phi$ along each dimension. Along the x-axis, the probability can therefore be calculated as:

$$P_{cx}\left(\mu' - \frac{R}{2} < r_x \le \mu' + \frac{R}{2}\right) = \Phi\left(\frac{r_x - \mu'_x + \frac{R}{2}}{\sigma_{dx}}\right) - \Phi\left(\frac{r_x - \mu'_x - \frac{R}{2}}{\sigma_{dx}}\right) \tag{20}$$

The dependence on navigation and surveillance performance is explicitly included as:

$$P_{cx}(\mu'_x - \frac{R}{2} < r_x \le \mu'_x + \frac{R}{2}) = \Phi\left(\frac{r_x - \mu'_x + \frac{R}{2}}{\sigma_{nav,x} + \sigma_{sur,x}}\right) - \Phi\left(\frac{r_x - \mu'_x - \frac{R}{2}}{\sigma_{nav,x} + \sigma_{sur,x}}\right) \tag{21}$$

The probabilities along the $y$ and $z$ axes are expressed in a similar form. Assuming that the probability of overlap in each dimension is independent, the total collision risk is then given by:

$$P_c = P_{cx}\,P_{cy}\,P_{cz} \tag{22}$$

*2.7. CNS Risk Volume*

Further discussion is provided regarding the construction of the volume around each aircraft. Previously, it was discussed that the combined covariance matrix $P$ is placed at the nominal location of the intruder aircraft. The individual components $P_a$ and $P_b$ of this volume can be inflated to provide a larger buffer zone around the aircraft commensurate to the target level of risk for a given mission. In this way, by designing trajectories such that the volume is not breached at any given epoch, it can be ensured that the target level of risk is not exceeded. The volume is, therefore, a risk volume whose avoidance protects the surrounding traffic against possible collisions. This is notionally depicted in Figure 15. The primary contributions to this volume are the navigation uncertainty and the surveillance uncertainty.

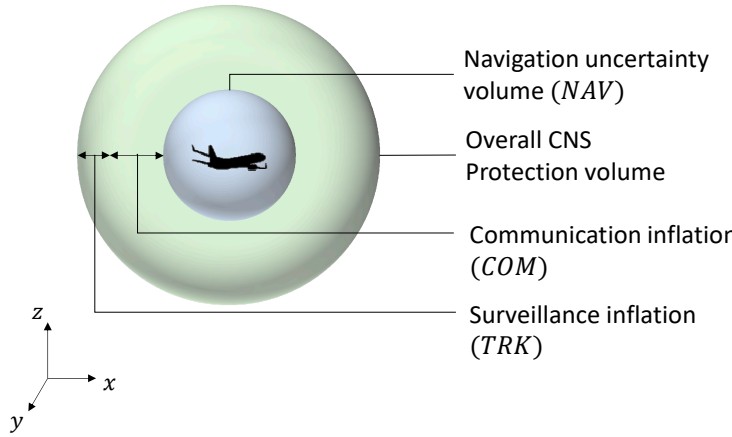

**Figure 15.** Conceptual depiction of the total protection volume as a result of all CNS factors.

Further inflations are added to the protection volume to account for additional failure modes potentially leading to a collision. For instance, an additional buffer to compensate for delays in receiving, acknowledging and enacting an instruction due to the limitations of the employed communication system(s) and delays introduced by operator/pilot response times is also depicted in Figure 15. In this manner, each system failure mode is translated to a spatial bound to form the overall protection volume.

In this study, the discussion is restricted to the failure of CNS systems. In this context, the cause of a collision can be traced back to a failure of one or more of the following elements:

- Failure of navigation systems: loss of accuracy beyond a specified limit without timely detection. This constitutes hazardously misleading information preventing timely recovery action from pilots/controllers;
- Failure of surveillance systems: loss in accuracy of aircraft localisation and non-timely relay of surveillance information to downstream sub-systems for recovery actions;
- Failure of Communication: loss/degradation of a link to the point where necessary recovery actions cannot be implemented in a timely manner.

As previously mentioned, the framework also allows the addition of further inflations to account for other factors. Two of these are mentioned here:

- Relative dynamics: an inflation is introduced, which is commensurate with the known or assumed closure rate of the aircraft;
- Wake turbulence: a buffer region is added, which guarantees sufficient separation from the hazardous region in the wake of the preceding aircraft.

The approach we follow in this paper is to generate protection volumes and allocate safety buffers based on theoretical models (with conservative assumptions) of system performance and environmental factors. For operational deployment, these theoretical bounds can be reduced based on actual air traffic data gathered in a certain UTM scenario and over extended time periods. Regarding the inflation associated with relative dynamics, this can either be directional or omnidirectional. For instance, in a hypothetical worst-case scenario where only non-cooperative surveillance is available, the estimation of the velocity vector (absolute or relative) of an intruder aircraft would likely be very inaccurate, especially in the early detection stages. In this case, the most conservative approach assumes that the direction and magnitude of such a velocity vector can take any value that is theoretically possible [36]. In our unified collision risk model, this mathematically corresponds to omnidirectional inflation of the original protection volume, of a magnitude sufficient to encompass the minimum and maximum flight speeds which are theoretically possible.

The potential failure causes and their combinatorial logic leading to a collision for a specific scenario are illustrated in Figure 16.

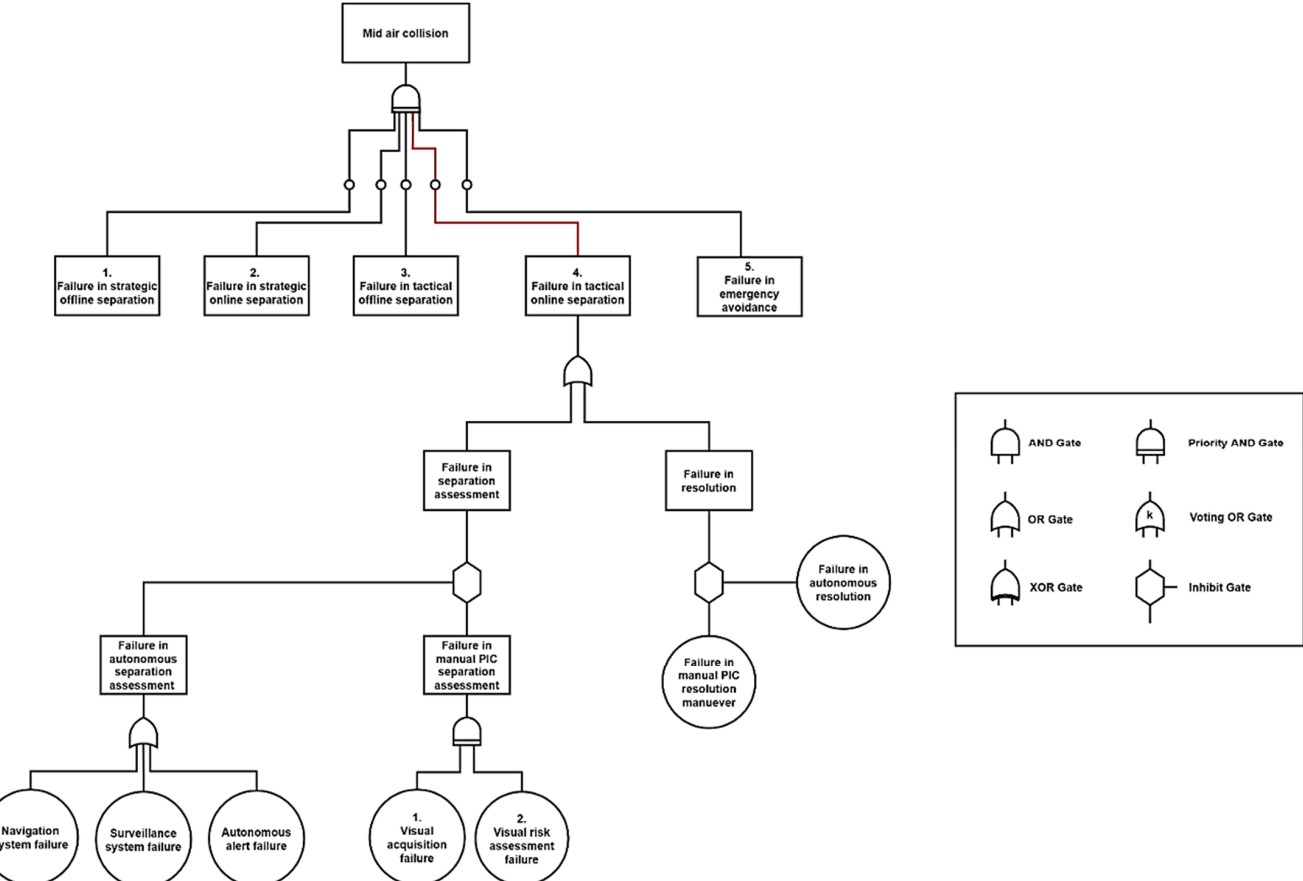

**Figure 16.** Sequential combination of system failures leading to a collision. The failure in tactical online separation is expanded.

Separation in this instance is provided through multiple sequential layers of protection in the strategic and tactical timeframes. This is captured through the use of a priority AND gate for which the output is true if the inputs are all true in a specified sequence. The sequence of failures is denoted within the block (i.e., tactical failures are followed by strategic failures). The faults leading to a failure in tactical online separation are expanded.

In this instance, a loss of separation is the consequence of either a failure to assess potential collision risk or a failure to execute a timely resolution manoeuvre. Both the assessment and resolution are performed primarily by onboard autonomous functionalities. A failure of these functionalities leads to fully manual separation assurance actions performed by the Pilot in Charge (PIC). The fault combinatorial logic can be represented by a combination of basic and more sophisticated logic gates.

Each branch emerging from the topmost event are applicable to different collision scenarios depending on the available CNS elements and human operators involved to accomplish the Separation Assurance and Collision Avoidance (SA/CA) tasks. For instance, for VFR aircraft, a collision can occur due to a failure in self-separation as executed by the pilot(s). For IFR aircraft, a collision can occur as the result of a failure of airborne cooperative separation assurance or due to a failure in ground-based surveillance and ATC intervention systems (whichever is employed at a given point in time). Conceptually, each of the terminal events in the fault tree corresponds to an inflation of the original volume or an additional buffer layer. The dimensions of each are discussed next.

### 2.7.1. Risk Volume Decomposition

The risk volume comprises a number of layers:

- Navigation: The navigation component of the volume is conveniently represented by a Gaussian ellipsoid. For most avionics systems, navigation sensor outputs are typically fused in a state estimator such as a Kalman filter, which outputs an optimal estimate of aircraft position as well as position uncertainty in the form of a state covariance matrix. Alternately, a more standardised and conservative representation of position uncertainty, the protection level, can be utilised. A protection level is essentially an upper bound on the position error for a given navigation system, which also takes into account the employed fault detection and isolation algorithm. The methodology of inflating the protection level to bound errors to a target probability is also well defined.
- Surveillance: The error in localising the intruder aircraft arises from two sources. The first of these is the error arising from the sensor used to measure the states of the intruder. The second arises from the time difference between observing the intruder and utilising the observation to assess the likelihood of a threat. Depending on the type of employed system, there can be a significant component of latency in the surveillance system. If the intruder aircraft are observed through a cooperative surveillance system such as ADS-B, then the error in localising the aircraft error will essentially be dependent on the error of the onboard GNSS system [43,44]. If a non-cooperative system such as radar is used to observe the intruder, then the localisation error will depend on the deployed radar characteristics and the intruder parameters. Before the estimated position of the intruder is used to assess the likelihood of a collision, it is compensated for latency in the system. Since latency cannot be directly observed in real-time, latency compensation is based on a priori modelled or measured system characteristics. Therefore, a residual error component will remain.
- Communication: If the assessment of collision risk is performed on the ground, either through an autonomous system or through visual observation by the remote PIC, then the execution of an avoidance manoeuvre is performed through the communication link between the GCS and the unmanned platform. The communication component of the risk volume is determined to provide a sufficient buffer that protects against a loss of separation due to a failure of the communication system.

The mathematical models underpinning the dimensions of two of these buffers—navigation and surveillance, are covered in the next sub-section.

### 2.7.2. Navigation and Surveillance Modelling

The localisation error that must be accounted for in the risk volume generation is mathematically described in this sub-section. The scope of the navigation and surveil-

lance modelling in this paper is restricted to GNSS and Primary Surveillance Radar (PSR), respectively.

Primary Surveillance Radar Errors

Radar calculates the target location by measuring its range and two angular coordinates with respect to the radar position. The angular coordinates commonly used are the elevation angle that is relative to the local horizontal and the azimuth measured relative to the true north [45]. A tracking radar has to initially identify the target in space and then determine its range and angular coordinates [46]. Curry [45] summarises the multiple sources that can cause errors in the radar target measurement, such as:

- A Signal to Noise (S/N) dependent random measurement error
- Random measurement error with fixed standard deviation due to noise sources in the radar receiver's final stages. These errors are usually small and correspond to the S/N dependent errors that are produced when S/N is high
- A bias error that occurs due to radar calibration and measurement
- Errors due to conditions of radar propagation and the uncertainties in correcting these errors
- Interference errors that occur due to various reasons such as radar clutter

A range measurement error equation is determined based on the major causes of range error [45]:

$$\sigma_R = \left( \sigma_{RN}^2 + \sigma_{RF}^2 + \sigma_{RB}^2 \right)^{1/2} \tag{23}$$

where $\sigma_{RN}$ is the S/N dependent random range measurement error, $\sigma_{RN} = \frac{\Delta R}{\sqrt{2\left(\frac{S}{N}\right)}}$ S/N is the signal-to-noise ratio, and $\Delta R$ is the Radar Range Resolution ($\Delta R = \frac{c}{2B}$, $c$ is the speed of sound and $B$ is the bandwidth of the signal)

$\sigma_{RF}$ is the random error with a fixed standard deviation produced when S/N is high. $\sigma_{RB}$ is the range bias error as a result of calibration and measurement.

Tracking refers to a radar identifying the position of one or more objects in space. Errors also occur when determining the target's azimuth and elevation angles and the requirements determining the accuracy of these angle measurements for a tracking radar are more exacting than those for a search radar [46]. An angular measurement error equation is based on the major causes of angular errors on a radar system:

$$\sigma_A = \left( \sigma_\omega^2 + \sigma_\varepsilon^2 + \sigma_{EBG}^2 \right)^{1/2} \tag{24}$$

where $\sigma_\omega$ is the rotational error of the radar system occurring due to the jitter of the motor mounted at the base of the radar that enables the rotation of the radar antenna; $\sigma_\varepsilon$ is the error in determining the elevation of the target; $\sigma_{EBG}$ is the error from the electron beam generated.

GNSS Errors

The main error sources affecting GNSS positioning are the pseudorange and carrier phase measurements. Sabatini et al. [47] summarise these errors into the following categories:

1. Receiver Dependent Errors such as Clock Error, Noise and Resolution;
2. Ephemeris Prediction Errors;
3. Satellite Dependent Errors that include Clock Offset and Group Delays;
4. Propagation Errors such as Ionospheric Delay, Tropospheric Delay and Multipath;
5. User Dynamics Error.

The User Equivalent Range Error (*UERE*) is a vector alongside the line-of-sight of the user-satellite that is a resultant of the projection of all system errors and is given by the following equation [47]:

$$UERE = \sqrt{\sigma_{e+cl}^2 + \sigma_{atm}^2 + \sigma_{mp}^2 + \sigma_n^2} \tag{25}$$

where $\sigma_{e+cl}$ is the broadcast ephemeris and clock error, $\sigma_{atm}$ is the atmospheric (ionospheric and tropospheric) error, $\sigma_{mp}$ is the multipath interference and $\sigma_n$ is the receiver noise.

The GNSS accuracy is not dependent on just the ranging errors, which determines position accuracy, but also on the navigation accuracy that is determined by the relative geometry of the satellites and the user and is given by the Dilution of Precision (DOP) factors [47].

Vertical Dilution of Precision (*VDOP*)

$$VDOP = \sigma_h \tag{26}$$

Horizontal Dilution of Precision (*HDOP*)

$$HDOP = \sqrt{\sigma_n^2 + \sigma_e^2} \tag{27}$$

Position Dilution of Precision (*PDOP*)

$$PDOP = \sqrt{\sigma_n^2 + \sigma_e^2 + \sigma_h^2} \tag{28}$$

Time Dilution of Precision (*TDOP*)

$$TDOP = \sigma_\tau \tag{29}$$

where $\sigma_e$, $\sigma_n$, $\sigma_h$ are the variances of east, north and height components, $\sigma_\tau$ is the standard deviation in the receiver clock bias. The Estimated Position Error (*EPE*) and the Estimated Time Errors (*ETE*) of a GNSS receiver can be calculated using the *PDOP* (contributing to EPE in 3D), the HDOP (contributing to *EPE* in 2D) and the *TDOP* and are given by the following equations [47]:

$$EPE_{3D} = \sigma_R \cdot PDOP \tag{30}$$

$$EPE_{2D} = \sigma_R \cdot HDOP \tag{31}$$

$$ETE = \sigma_R \cdot TDOP \tag{32}$$

where $\sigma_R$ is the standard deviation of pseudo-range measurement error. Several longstanding models exist for each component of the UERE [47]. However, many of these models will have to be adapted to the UAS context. For example, low-altitude UAS operations in an urban environment would be vulnerable to severe multipath, and traditional multipath models for aviation do not account for the specificities of the urban environment [43].

ADS-B Errors

ADS-B is a cooperative surveillance system wherein aircraft share data through a network. The relevant messages shared via the communication link include aircraft position, velocity, and ID, which are transmitted through a Mode-S Extended Squitter (1090 MHz). This type of scenario is illustrated in Figure 17.

As per aviation standards [48], the aircraft position and velocity is obtained from the onboard GNSS receiver. Therefore, the localisation component of the error associated with a given ADS-B system is essentially the error affecting the position estimate of the GNSS receiver (as modelled in Section GNSS Errors). In aviation, GNSS-based navigation systems are categorised based on their performance into distinct Required Navigation Performance (RNP) groups, which specify most importantly for the collision risk problem, the accuracy of the computed position solution and its integrity [49,50]. In addition to the localisation error, the latency between when the measurement is made and when it is utilised to assess risk must also be accounted for.

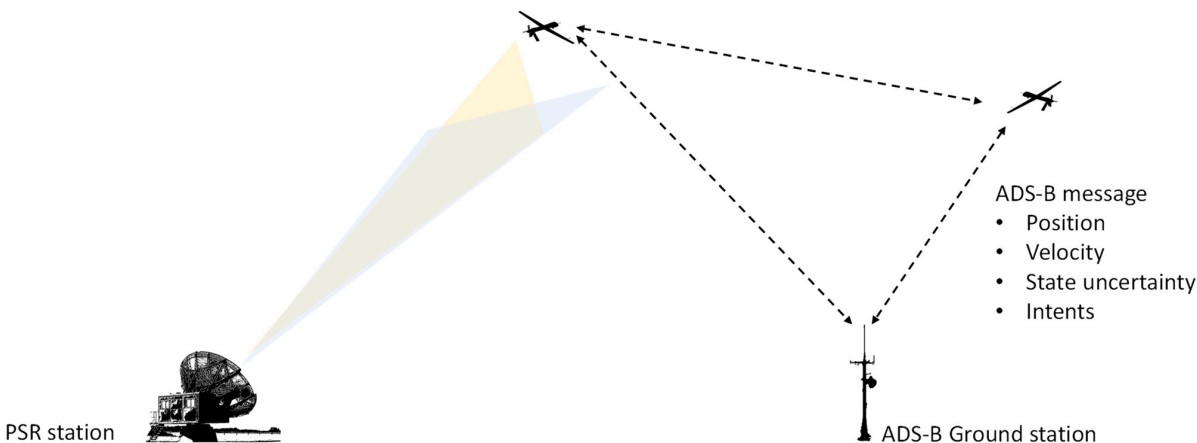

**Figure 17.** Air traffic surveillance modalities.

### 2.7.3. Volume Coordinate Transformation

To combine the different layers into a unified risk volume, the layers must be described in a common coordinate system. Assumed uncertainties for navigation and tracking based on GNSS and Radar are defined as $[\sigma_{X\_GNSS}, \sigma_{Y\_GNSS}, \sigma_{Z\_GNSS}]$ and $[\sigma_{R\_Radar}, \sigma_{A\_Radar}, \sigma_{E\_Radar}]$ respectively. The tracking uncertainty ellipsoid resulting from the azimuth, elevation and range errors is given by the following equations:

$$R_{radar} = R_T + \sigma_R cos\eta \, cos\nu \tag{33}$$

$$\alpha_{radar} = \alpha_T + \sigma_\alpha sin\eta \, cos\nu \tag{34}$$

$$\varepsilon_{radar} = \varepsilon_T + \sigma_\varepsilon sin\nu \tag{35}$$

where $r_T$, $\alpha_T$ and $\varepsilon_T$ is the position of the detected target in terms of range, azimuth and elevation. The practice employed in this paper will be to transform the radar observations of the host and intruder aircraft into a local Cartesian coordinate system.

Transformation of this ellipsoid is given by the following equations

$$X_{T\_radar} = X_{radar} + R_{radar}cos\alpha_{radar}cos\varepsilon_{radar} \tag{36}$$

$$Y_{T\_radar} = Y_{radar} + R_{radar}sin\alpha_{radar}cos\varepsilon_{radar} \tag{37}$$

$$Z_{T\_radar} = Z_{radar} + R_{radar}sin\varepsilon_{radar} \tag{38}$$

Substituting $R_{radar}$, $\alpha_{radar}$, $\varepsilon_{radar}$ from the equations above

$$X_{T\_radar} = X_{radar} + (R_T + \sigma_R cos\eta cos\nu) \times \cos(\alpha_T + \sigma_\alpha sin\eta cos\nu) \times \cos(\varepsilon_T + \sigma_\varepsilon sin\nu) \tag{39}$$

$$Y_{T\_radar} = Y_{radar} + (R_T + \sigma_R cos\eta cos\nu) \times \sin(\alpha_T + \sigma_\alpha sin\eta cos\nu) \times \cos(\varepsilon_T + \sigma_\varepsilon sin\nu) \tag{40}$$

$$Z_{T\_radar} = Z_{radar} + (R_T + \sigma_R cos\eta cos\nu) \times \sin(\varepsilon_T + \sigma_\varepsilon sin\nu) \tag{41}$$

Expanding the above equations:

$$
\begin{aligned}
X_{T\_radar} = {} & X_{radar} + (R_T + \sigma_R cos\eta \, cos\nu)[\cos\alpha_T \, \cos(\sigma_\alpha sin\eta \, cos\nu) \, \cos\varepsilon_T \cos(\sigma_\varepsilon sin\nu) \\
& - \cos\alpha_T \, \cos(\sigma_\alpha sin\eta \, cos\nu) \, \sin\varepsilon_T \sin(\sigma_\varepsilon sin\nu) - \sin\alpha_T \sin(\sigma_\alpha sin\eta \, cos\nu) \, \cos\varepsilon_T \cos(\sigma_\varepsilon sin\nu) \\
& + \sin\alpha_T \sin(\sigma_\alpha sin\eta \, cos\nu) \sin\varepsilon_T \sin(\sigma_\varepsilon sin\nu)]
\end{aligned}
\tag{42}
$$

$$
\begin{aligned}
Y_{T\_radar} = {} & Y_{radar} + (R_T + \sigma_R \cos\eta \cos\nu)[\sin\alpha_T \, \cos(\sigma_\alpha \sin\eta \cos\nu) \cos\varepsilon_T \cos(\sigma_\varepsilon \sin\nu) \\
& - \sin\alpha_T \cos(\sigma_\alpha \sin\eta \cos\nu) \, \sin\varepsilon_T \sin(\sigma_\varepsilon \sin\nu) + \cos\alpha_T \sin(\sigma_\alpha \sin\eta \cos\nu) \, \cos\varepsilon_T \cos(\sigma_\varepsilon \sin\nu) \\
& - \cos\alpha_T \sin(\sigma_\alpha \sin\eta \cos\nu) \sin\varepsilon_T \sin(\sigma_\varepsilon \sin\nu)]
\end{aligned}
\tag{43}
$$

$$Z_{T\_radar} = Z_{radar} + (R_T + \sigma_R \cos\eta \cos\nu)[\sin\varepsilon_T \cos(\sigma_\varepsilon \sin\nu) - \cos\varepsilon_T \sin(\sigma_\varepsilon \sin\nu)] \tag{44}$$

Mean error of the tracking error ellipsoid is given by the following equations:

$$\mu_{x,Radar} = X_{radar} + \mu_{R,Radar}\cos\mu_{\alpha,Radar}\cos\mu_{\varepsilon,Radar} \tag{45}$$

$$\mu_{y,Radar} = Y_{radar} + \mu_{R,Radar}\sin\mu_{\alpha,Radar}\cos\mu_{\varepsilon,Radar} \tag{46}$$

$$\mu_{z,Radar} = Z_{radar} + \mu_{R,Radar}\sin\mu_{\varepsilon,Radar} \tag{47}$$

Assuming the range, azimuth and elevation of the target tracked by the radar as $r_T$, $\alpha_T$, $\varepsilon_T$ respectively, the tracking error uncertainty ellipsoid is given by the following equations:

$$\begin{aligned}
\sigma_{X\_ellipse\_Radar} =\ & X_{radar} \\
& + (R_T + \sigma_R\cos\eta\cos\nu)[\cos\alpha_T\ \cos(\sigma_\alpha\sin\eta\cos\nu)\ \cos\varepsilon_T\cos(\sigma_\varepsilon\sin\nu) \\
& - \cos\alpha_T\ \cos(\sigma_\alpha\sin\eta\cos\nu)\cdot\sin\varepsilon_T\sin(\sigma_\varepsilon\sin\nu) - \sin\alpha_T\sin(\sigma_\alpha\sin\eta\cos\nu)\ \cos\varepsilon_T\cos(\sigma_\varepsilon\sin\nu) \\
& + \sin\alpha_T\sin(\sigma_\alpha\sin\eta\cos\nu)\sin\varepsilon_T\sin(\sigma_\varepsilon\sin\nu)] - X_{radar} - r_T\cos\alpha_T\cos\varepsilon_T
\end{aligned} \tag{48}$$

$$\begin{aligned}
\sigma_{Y\_ellipse\_Radar} =\ & Y_{radar} \\
& + (R_T + \sigma_R\cos\eta\cos\nu)[\sin\alpha_T\ \cos(\sigma_\alpha\sin\eta\cos\nu)\ \cos\varepsilon_T\cos(\sigma_\varepsilon\sin\nu) \\
& - \sin\alpha_T\ \cos(\sigma_\alpha\sin\eta\cos\nu)\ \sin\varepsilon_T\sin(\sigma_\varepsilon\sin\nu) + \cos\alpha_T\sin(\sigma_\alpha\sin\eta\cos\nu)\ \cos\varepsilon_T\cos(\sigma_\varepsilon\sin\nu) \\
& - \cos\alpha_T\sin(\sigma_\alpha sin\eta cos\nu)\sin\varepsilon_T\sin(\sigma_\varepsilon sin\nu)] - Y_{radar} - r_T\sin\alpha_T\cos\varepsilon_T
\end{aligned} \tag{49}$$

$$\sigma_{Z_{ellipse_{Radar}}} = Z_{radar} + (R_T + \sigma_R cos\eta cos\nu)[\sin\varepsilon_T\cos(\sigma_\varepsilon sin\nu) - \cos\varepsilon_T\sin(\sigma_\varepsilon sin\nu)]Z_{radar} - r_T sin\varepsilon_T \tag{50}$$

Since the rotation transform is non-linear, the Jacobian of the rotation matrix is used. This locally linear approximation of the rotation is performed to preserve the Gaussianity of the tracking errors. Considering a GNSS-based navigation system, the navigation uncertainty ellipsoid is given by the following equations, which are natively in the Cartesian frame:

$$\sigma_{X\_ellipse\_GNSS} = \sigma_{X\_GNSS}cos\eta\ cos\nu \tag{51}$$

$$\sigma_{Y\_ellipse\_GNSS} = \sigma_{Y\_GNSS}sin\eta\ cos\nu \tag{52}$$

$$\sigma_{Z\_ellipse\_GNSS} = \sigma_{Z\_GNSS}sin\nu \tag{53}$$

Examples of uncertainty ellipsoids due to Radar, GNSS and total system error are plotted in Figure 18. The radar uncertainty volume is transformed into a Cartesian frame and combined with the GNSS uncertainty volume as described in [51].

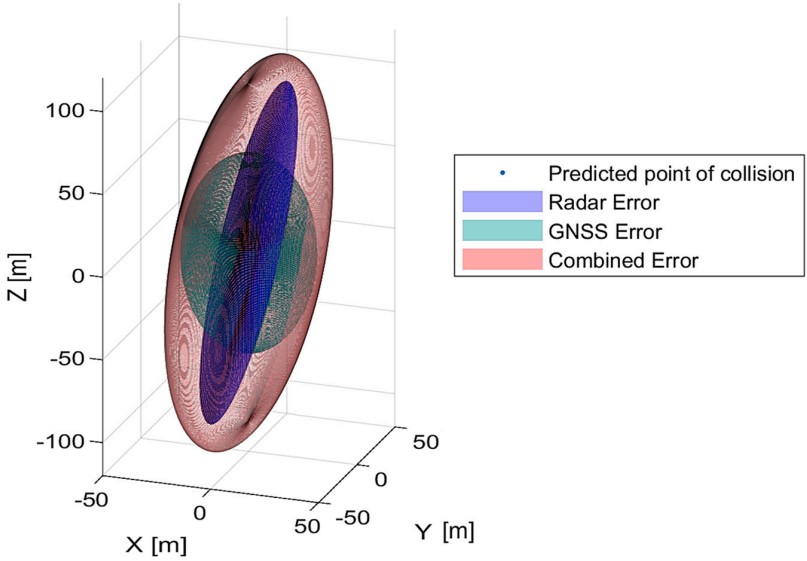

**Figure 18.** Combined GNSS and Radar Errors.

## 3. Application Case Studies

Two case studies are presented in this section to illustrate the application of the proposed CNS performance-based models in different contexts and to analyse the related results.

### 3.1. Terminal Control Area

A UAS encounter in a terminal area scenario is analysed upfront. The terminal area scenario specifications are provided in Table 1. In the terminal area scenario, both aircraft are tracked by a single ground-based primary surveillance radar. The two aircraft are in level flight at the same altitude and are on intersecting trajectories, as illustrated in Figure 19. The collision risk is plotted in Figure 20. The radar errors of the two aircraft are correlated as they are being tracked by the same radar platform. In the event that two aircraft are being tracked by the same radar, there exists a covariance between the tracking uncertainties of both aircraft that needs to be determined and encapsulated in the error volume modelling.

**Table 1.** Terminal area scenario specifications.

| Operational Factors | Specifications |
|---|---|
| Aircraft | Fixed wing UAVs<br>Wingspan: 15 m |
| Surveillance | Monostatic scanning PSR<br>$\sigma_R$ = 10 m<br>$\sigma_{Az}$ = 13<br>$\sigma_{El}$ = 26 |
| Trajectories | Level flight (altitude: 110 m AGL)<br>Constant speed (25 m/s) and heading<br>Intersecting routes |

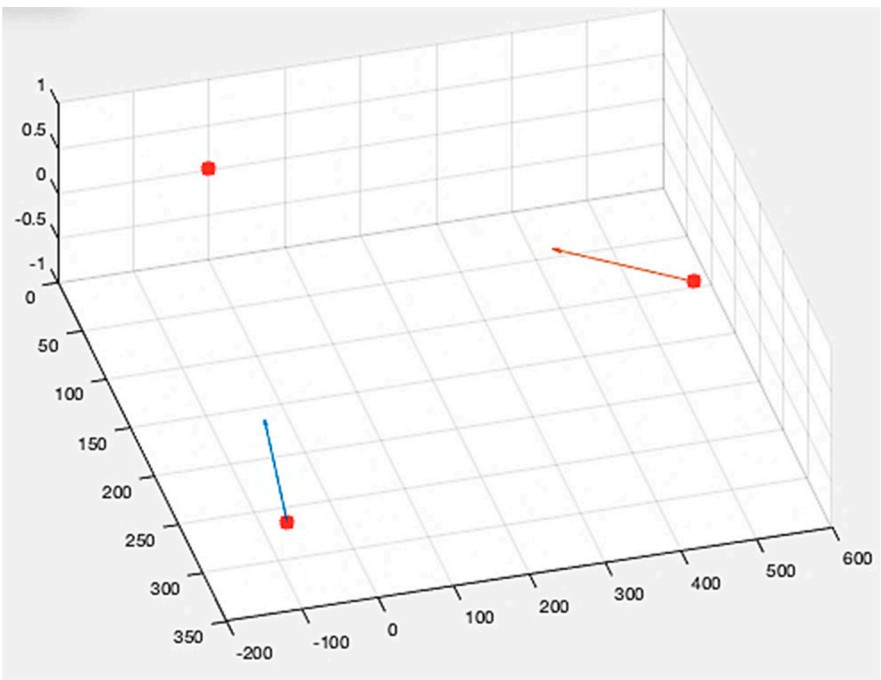

**Figure 19.** Terminal area scenario—two aircraft on intersecting trajectories.

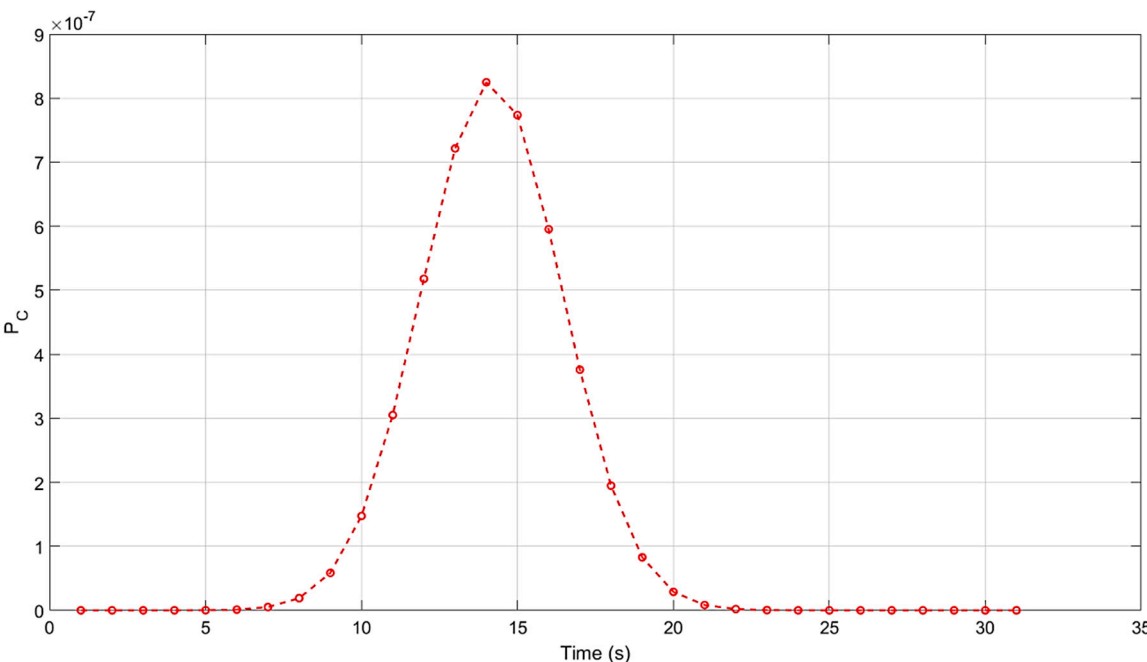

**Figure 20.** Collision risk for the intersecting trajectories scenario.

### 3.2. Enroute

A set of UAS encounters in an enroute scenario was also simulated. The specifications of this enroute scenario are summarised in Table 2. The surveillance system considered in this case is ADS-B. The two aircraft follow parallel and level routes with constant speed and heading. The risk incurred is examined for two navigation accuracy categories: (1) $2\sigma$ = 185.2 m; $2\sigma$ = 75 m. The first category corresponds to an RNP 0.1 accuracy specification, whereas the second category corresponds to a containment radius of 75 m (a notional RNP 0.04 specification).

**Table 2.** Enroute area scenario specifications.

| Operational Factors | Specifications |
|---|---|
| Aircraft | Fixed wing UAVs<br>Wingspan: 15 m |
| Surveillance | ADS-B |
| Navigation | GNSS receiver: GPS<br>RNP 0.1; Accuracy: 185.2 m<br>RNP 0.04; Accuracy: 75 m |
| Trajectories | Level flight (altitude: 110 m/s AGL)<br>Constant speed (25 m/s, 40 m/s) and heading<br>Parallel routes |

The navigation performance is modelled using a GNSS constellation and receiver simulator. An elevation mask of $10°$ is applied to extract visible satellites. Each receiver has an EKF-based processor for estimating position and velocity from the satellite-to-receiver pseudoranges. Each aircraft is assumed to track the other using ADS-B transceivers.

The collision risks time series calculated for the abovementioned enroute cases are shown in Figure 21. As expected, the risk incurred is an order of magnitude lower in the case of a higher navigation performance standard for a given separation distance—$P_c = 2.9 \times 10^{-4}$ for RNP 0.1 and $P_c = 1.19 \times 10^{-6}$ for the navigation containment of 75 m. The risk exposure is also greater in the case of an overtaking scenario owing to the longer duration of the encounter.

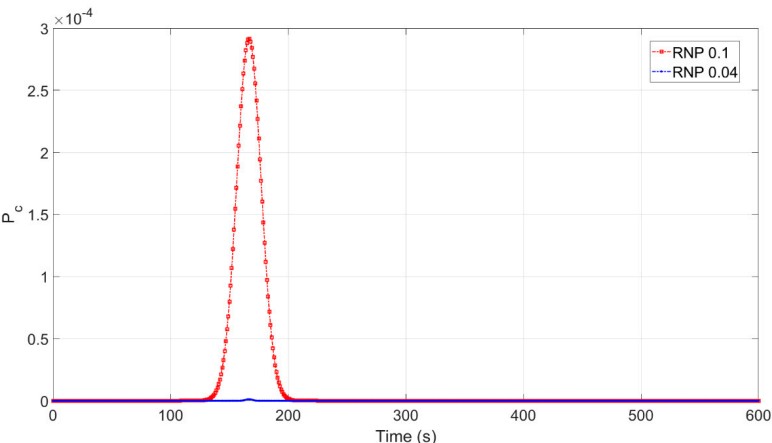

**Figure 21.** Collision risk results for the enroute scenario.

*3.3. Discussion*

In general, an inverse relationship between CNS performance and collision risk is demonstrated. Although only navigation performance from current RNP standards was investigated in this instance, the results are generalisable to surveillance and communication performance as well. To investigate this trade-off more closely, the collision risk for the parallel trajectory scenario was computed for different combinations of horizontal navigation performance and trajectory separation thresholds. Assuming both aircraft are flying level at the same altitude, the model is used to assess the combinations of navigation accuracy (95% RMS) and trajectory spacing. The collision probability at the closest point of approach is evaluated for each of these combinations. Navigation accuracy is varied up to 500 m (i.e., a value between RNP 0.1 and 0.3. Surveillance performance is set at a constant value). An ADS-B performance of 200 m was assumed (i.e., the intruder can be localised within a containment bound of a radius of 200 m). This corresponds to the NIC 8 performance category for ADS-B. Trajectory separation is varied from 1000 m to 2000 m. The risk trade-off is illustrated in Figure 22. As expected, collision risk increases with a lower separation and lower navigation accuracy. Given the maximum achievable accuracy by the navigation equipment, the required separation can be evaluated or inversely, the required accuracy can be determined to meet a maximum allowable separation due to airspace constraints. The lowest recorded risk was $2 \times 10^{-6}$ at a separation slightly greater than 1200 m for a horizontal accuracy of 280 m. Assuming this number to be the limit of achievable navigation performance, risk reduction below this value would then require either greater separation or improved surveillance performance.

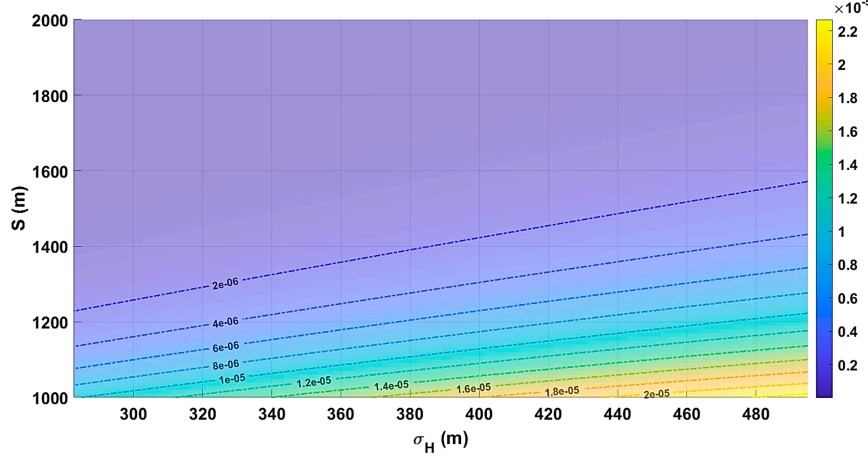

**Figure 22.** Navigation performance trade-off against trajectory separation. 95% navigation accuracy bound versus trajectory spacing.

## 4. Conclusions

This article presented a performance-based collision risk model and associated airspace risk assessment framework, which address some of the key limitations of pre-existing approaches. The unified collision risk model accounts for the varying Communication, Navigation and Surveillance (CNS) infrastructure and system mandates that apply to different sectors of the conventional airspace and, potentially, to future low-altitude and very high-altitude airspace (i.e., below and above current controlled airspace). The model is specifically tailored to facilitate the integration of Unmanned Aircraft Systems (UAS) and Urban Air Mobility (UAM) in unsegregated airspace and to address the complexities associated with such evolution. In addition to CNS performance, factors such as relative dynamics, adverse weather and human decision-making processes can be easily incorporated into the model and translated to conservative spatial bounds, allowing the computation of convenient protection volumes around each aircraft and ground obstacle in a given scenario, which are based on statistical modelling of all factors involved in the predicted encounters. Additionally, an efficient method for approximating the probability of collision was introduced, which conservatively evaluates horizontal and vertical collision probabilities. Such conservative approximation can be analytically inverted to inform the design of an airspace region starting from a preselected level of acceptable risk. The model's applicability to strategic offline scenarios was demonstrated through two case studies, where Primary Surveillance Radar (PSR) and Automatic Dependent Surveillance-Broadcast (ADS-B) were separately considered. The sensitivity of collision risk to navigation performance was a core element in these case studies, aiming to demonstrate the effectiveness of the proposed CNS performance-based formulation. A maximum risk of $1.1 \times 10^{-6}$ was determined for the intersecting route scenario. In the parallel route scenario, a $2\sigma$ navigation accuracy bound of 185.2 m was found to incur a maximum collision risk of approximately $1.5 \times 10^{-4}$ at a separation of 200 m. By reducing the required navigation accuracy bound to 92.6 m, a risk of approximately $0.1 \times 10^{-4}$ was incurred, representing a reduction of approximately 93%. Future work will extend the analytical framework to include the impacts of communication performance and relative dynamics on the collision risk. Additionally, the potential of incorporating factors such as adverse weather and wind gusts as additional buffers in the protection volume will be investigated. Particular focus will be on the extension of the model to capture the notable differences in manoeuvrability, guidance, path-planning and wake turbulence characteristics of Electric Vertical Take-Off and Landing (eVTOL) and various other emerging rotary-wing platforms compared to fixed-wing aircraft. This development will necessarily occur after the conclusion of various ongoing studies in the dynamics modelling and handling qualities of these new aircraft [52–55]. Future work will also address the application of the proposed collision risk model to tactical online ATM operations and online risk estimation performed onboard UAS platforms. This will include the investigation of different numerical methods for evaluating risk in real-time/near real-time and the corresponding trade-off in evaluation accuracy. This line of future work will also explore the synergy between different online system performance estimation methods and risk estimation models. Another key focus area will be on the application of risk mitigation and control strategies suitable for 4-Dimensional Trajectory Optimisation (4DTO) and Dynamic Airspace Management (DAM). The application of the model to assess risk under different GNSS augmentation strategies will also be studied. These include existing Space Based Augmentation Systems (SBAS) and the future Dual Frequency Multi-Constellation (DFMC) SBAS. Another promising area of future work is the use of the model to assess the impact of airspace design on risk for AAM platforms.

**Author Contributions:** Conceptualisation, R.S., A.G., S.B. and N.P.; methodology, R.S., S.B. and A.G.; software, S.B. and A.G.; writing—original draft preparation, S.B., A.G. and N.P.; writing—review and editing, R.S. and T.K.; project administration, R.S. and A.G.; funding acquisition, R.S. All authors have read and agreed to the published version of the manuscript.

**Funding:** Portions of this research were supported by the Australian Civil Aviation Safety Authority (CASA) under contract no. 19/39—Airspace Risk Modelling Research Program (ARM-RP).

**Institutional Review Board Statement:** Not applicable.

**Informed Consent Statement:** Not applicable.

**Data Availability Statement:** Data will be made available upon request to the corresponding author.

**Acknowledgments:** Parts of the study reported in this paper are derived from an Airspace Risk Modelling research report prepared by RMIT for the Civil Aviation Safety Authority (CASA) in 2020. The contents of this paper do not necessarily reflect the views of CASA.

**Conflicts of Interest:** The authors declare no conflict of interest. The funding bodies had no role in the design of this study, in the collection, analysis or interpretation of the data, in the writing of the manuscript or in the decision to publish the results.

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
