# Peer review of "A Unified Airspace Risk Management Framework for UAS Operations†"

_drones, doi:10.3390/drones6070184_

Round 1

Reviewer 1 Report

Your paper is particularly good and is a good update for those studying the implementation of UTMs. The literature review is very comprehensive and well documented. Your framework is a fresh update into the study of multi-agent implementation of drones in airways. It was interesting that you implemented the gas particle theories within the text. There are still some more collision avoidance models published that you might like to investigate. There are other models that identify the risk within the type of population under the drone and weather conditions. The mathematical model proposed covers multiple important variables and does show promise. However, its implementation is more for the ATM or operator level. More for the all-knowing observer  (centralize vs. decentralized) that coordinates the traffic. It is also hard to see something so complex can be practical to implement within a drone itself. Could you program your mathematical model so the drone itself could calculate the risk? Or the operator needs to update the risk for it? And a multi-agent implementation seems very computationally demanding your model.

Other thing that must be accounting moving forward is the study of validation or risk created by autonomy. Motion planning and path planning algorithms do not seem to be accounting in your model and they are essential into fully automating UAS. The plan is to have these devices flying themselves without an operator. It will be interesting to see how your framework evolves by implementing autonomy, machine learning and the possibilities of fuzzy logic as tools to enhance drones’ capabilities for collision avoidance. 

Also, your model follows the perspective of a plane model and does not seem to add into the enhance mobility of a multi-copter or hybrid drone. The hovering capability itself, is an immensely powerful risk mitigation parameter beyond of what a fixed-wing drone can do. I would like to see more work into that on your future work

The only correction needed is in line 503, in which the number for equation 20 only has a zero without a two before. 

The paper as-is should be accepted. I do not detect any needed major changes. Just keep moving forward and keep on enhancing your framework.

Author Response

Reviewer #1

Comment

Action

Your paper is particularly good and is a good update for those studying the implementation of UTMs. The literature review is very comprehensive and well documented. Your framework is a fresh update into the study of multi-agent implementation of drones in airways. It was interesting that you implemented the gas particle theories within the text. There are still some more collision avoidance models published that you might like to investigate. There are other models that identify the risk within the type of population under the drone and weather conditions. The mathematical model proposed covers multiple important variables and does show promise. However, its implementation is more for the ATM or operator level. More for the all-knowing observer (centralize vs. decentralized) that coordinates the traffic. It is also hard to see something so complex can be practical to implement within a drone itself. Could you program your mathematical model so the drone itself could calculate the risk? Or the operator needs to update the risk for it? And a multi-agent implementation seems very computationally demanding your model.

The authors thank the reviewer for the valuable feedback. We have highlighted the scope of the paper to focus on the strategic timeframe. We have also added further discussion on future work in the conclusion in which we mention our intent to extend the model to online risk evaluation performed onboard the platform.

Also, your model follows the perspective of a plane model and does not seem to add into the enhance mobility of a multi-copter or hybrid drone. The hovering capability itself, is an immensely powerful risk mitigation parameter beyond of what a fixed-wing drone can do. I would like to see more work into that on your future work

The authors thank the reviewer for the valuable feedback. we have now also highlighted the importance to extend the proposed model to rotary-wing platforms in the recommendations for future research.

The only correction needed is in line 503, in which the number for equation 20 only has a zero without a two before.

This has been corrected. Thank you for pointing this out.

Reviewer 2 Report

In this paper, the authors proposed a comprehensive risk management framework to model UAS collision risk in all classes of airspace. This methodology inherently accounts for the performance of Communication, Navigation and Surveillance (CNS) systems. Additionally, the proposed approach can be applied inversely to determine CNS performance requirements given a target value of collision probability.

In general, this paper is organized and well written. I recommend this paper to be published.

The Figures in this paper should be improved in final version. The fonts in the figures are not standardized.

Author Response

Reviewer #2

Comment

Action

In general, this paper is organized and well written. I recommend this paper to be published.

The Figures in this paper should be improved in final version. The fonts in the figures are not standardized.

The authors thank the reviewer for the valuable feedback. We have corrected the figures and sharpened some of the figures as necessary.